 eLIFE

# Interplay between population firing stability and single neuron dynamics in hippocampal networks

**Edden Slomowitz[1†], Boaz Styr[1†], Irena Vertkin[1], Hila Milshtein-Parush[1,2], Israel Nelken[3,4], Michael Slutsky[5], Inna Slutsky[1,2]\***

[1]Department of Physiology and Pharmacology, Sackler Faculty of Medicine, Tel Aviv University, Tel Aviv, Israel; [2]Sagol School of Neuroscience, Tel Aviv University, Tel Aviv, Israel; [3]Department of Neurobiology, Alexander Silberman Institute of Life Sciences, Hebrew University, Jerusalem, Israel; [4]Edmond and Lily Safra Center for Brain Sciences, Hebrew University, Jerusalem, Israel; [5]Mantis Vision, Kiryat Arie, Israel

**Abstract** Neuronal circuits' ability to maintain the delicate balance between stability and flexibility in changing environments is critical for normal neuronal functioning. However, to what extent individual neurons and neuronal populations maintain internal firing properties remains largely unknown. In this study, we show that distributions of spontaneous population firing rates and synchrony are subject to accurate homeostatic control following increase of synaptic inhibition in cultured hippocampal networks. Reduction in firing rate triggered synaptic and intrinsic adaptive responses operating as global homeostatic mechanisms to maintain firing macro-stability, without achieving local homeostasis at the single-neuron level. Adaptive mechanisms, while stabilizing population firing properties, reduced short-term facilitation essential for synaptic discrimination of input patterns. Thus, invariant ongoing population dynamics emerge from intrinsically unstable activity patterns of individual neurons and synapses. The observed differences in the precision of homeostatic control at different spatial scales challenge cell-autonomous theory of network homeostasis and suggest the existence of network-wide regulation rules.

**\*For correspondence:** islutsky@post.tau.ac.il

[†]These authors contributed equally to this work

**Competing interests:** The authors declare that no competing interests exist.

## Introduction

Neural circuits achieve an ongoing balance between flexibility and stability to enable plastic adaptations to environmental changes, while maintaining neuronal activity in a stable regime over extended timescales. The balance between stability and plasticity has rarely been addressed in the past due to the technical challenge of monitoring the activity of the same neurons over extended timescales (*Lütcke et al., 2013*). Recently, great efforts have been made to establish a system for the monitoring of neuronal activity at long timescales in the hippocampus of freely moving mice (*Ziv et al., 2013*). That study, performed by monitoring $Ca^{2+}$ dynamics in thousands of CA1 cells over weeks through a miniature head-mounted microscope, revealed a remarkable degree of instability in the coding of space: only 25% of cells with place fields at one recording session exhibited the same properties 5 days later. Importantly, the diverse activity patterns at the single-neuron level gave rise to largely invariant spatial representations at the population level. Such results raise the question of whether inter-neuronal dynamics of spiking patterns stems from intrinsic variability of the hippocampal network and its constituent neurons, or whether they reflect extrinsic changes in hippocampal modulation by higher-order brain structures, adaptations to environmental changes, or subtle changes in animal behavior states.

Diverse homeostatic negative feedback systems operate to stabilize ongoing spiking properties in neuronal populations around a predefined 'set point' in face of constant environmental changes

**eLife digest** The human brain contains more than 80 billion neurons, which are organised into extensive networks. Changes in the strength of connections between neurons are thought to underlie learning and memory: neuronal networks must therefore be sufficiently stable to allow existing memories to be stored, while remaining flexible enough to enable the brain to form new memories.

Evidence suggests that the stability of neuronal networks is maintained by a process called homeostasis. If properties of the network—such as the average firing rate of all the neurons—deviate from a set point, changes occur to return the network the original set point. However, much less is known about the effects of homeostasis at the level of individual neurons within networks: do their firing rates also remain stable over time?

Slomowitz, Styr et al. have now addressed this question by recording the activity of neuronal networks grown on an array of electrodes. Applying a drug that inhibits neuronal firing caused the average firing rate of the networks to decrease initially, as expected. However, after 2 days, homeostasis had restored the average firing rate to its original value, despite the continued presence of the drug. By contrast, the individual neurons within the networks behaved differently: on day 2 almost 90% of neurons had a firing rate that was different from their original firing rate.

Similar behavior was seen when Slomowitz, Styr et al. studied the degree of synchronization between neurons as they fire: the average value for the network returned to its original value, but this did not happen at the level of individual neurons. Surprisingly, however, the ability of the network to undergo short-lived changes in average strength of the connections between neurons—which is thought to support short-term memory—was not subject to homeostasis. This suggests that the loss of short-term memory that occurs in many brain diseases may be an unfortunate consequence of the efforts of neuronal networks to keep their average responses stable.

(*Turrigiano and Nelson, 2004*; *Davis, 2006*; *Marder and Goaillard, 2006*). Extensive research lead to compelling evidence on a wide repertoire of possible homeostatic mechanisms, including adaptations of synaptic strength, changes in excitation–inhibition balance, and modulation of intrinsic excitability (*Turrigiano, 2011*). Despite progress in understanding the homeostatic mechanisms that underlie the stability of network firing properties, several key questions remain open. First, what are the basic properties of neural networks that are subjected to homeostatic control? Second, are homeostatic control systems equally precise at the level of individual neurons and neuronal populations? Third, what is the trigger of synaptic homeostatic mechanisms? And finally, how do compensatory changes in synaptic strength affect the network's functions?

To address these questions, long-term measurements of spiking activity over several days from the exact same neuron or population of neurons are required. While long-term monitoring of spiking activity cannot be reliably performed from the same neurons in vivo due to technical limitations, greater stability can be achieved in vitro. Therefore, we combined long-term extracellular spike recordings at the population and single-neuron levels using micro-electrode arrays (MEAs) and calcium imaging, together with intracellular patch-clamp measurements of synaptic responses and imaging of synaptic activity. We utilized cultured neuronal networks that represent an excellent experimental tool to study internal variability vs stability in a highly controlled environment. Here, we describe the basic relationships between ongoing spiking properties of individual neurons, population dynamics, and neuronal adaptive mechanisms.

## Results

### Quantifying GABA$_B$R-mediated inhibition at synaptic, single neuron, and network levels

While the majority of homeostatic mechanisms have been studied following complete pharmacological blockade of spikes or AMPAR-mediated excitatory postsynaptic currents (EPSCs), we decided to examine long-term effects of neuromodulation on firing properties of the network. Specifically, we examined how use-dependent synaptic inhibition via G-protein-coupled receptors, a widespread

mechanism of neuromodulation in the central nervous system (CNS), affects firing properties of the network over long timescales. As a perturbation, we chose a suppression of synaptic activity via widely expressed GABA$_B$ receptors (GABA$_B$Rs) using a selective GABA$_B$R agonist, baclofen. GABA$_B$Rs mediate presynaptic inhibition via inhibition of presynaptic calcium transients (**Wu and Saggau, 1995**; **Laviv et al., 2010**). Therefore, we first conducted experiments to determine the dose–response of baclofen at the level of presynaptic terminals utilizing FM1-43 dye in primary hippocampal cultures (**Abramov et al., 2009**). As GABA$_B$R-mediated inhibition is frequency-dependent (**Varela et al., 1997**; **Kreitzer and Regehr, 2000**; **Ohliger-Frerking et al., 2003**), we quantified the effects of baclofen for two types of input: low-frequency single spikes (30 stimuli at 1 Hz) and high-frequency spike bursts (30 stimuli delivered as 6 bursts; each burst contains 5 stimuli at 100 Hz), while maintaining the mean spiking rate constant at 1 Hz (**Figure 1A**). As expected, baclofen inhibits synaptic vesicle release, displaying preferential effects during low-frequency stimulation (**Figure 1A**). Quantitatively, IC$_{50}$ for single spikes was ~23-fold lower than for bursts (0.33 and 7.6 µM, respectively, **Figure 1B**). As a result, short-term facilitation was enhanced in a wide range of baclofen concentrations (0.01–10 µM, **Figure 1C**), indicating a conversion of synapses to a high-pass filter mode. Similar results were obtained using voltage-clamp experiments in acute hippocampal slices (**Figure 1—figure supplement 1A,B**).

Next, we examined how use-dependent blockade of synaptic transmission affects firing properties of hippocampal neurons and networks. For this purpose, we grew high-density hippocampal cultures on MEAs for ~3 weeks. Each MEA contains 59 recording electrodes with each electrode capable of recording the activity of several adjacent neurons (**Figure 1D**). Spikes were detected and analyzed using principal component analysis to obtain well-separated single units (**Figure 1E**). Unit firing rates were highly heterogeneous and skewed towards low frequencies (**Figure 1—figure supplement 1C**) as has been previously reported in vivo (**Mizuseki et al., 2012**). As expected, baclofen caused a fast and robust drop in single-unit firing rates (**Figure 1F,G**), leading to a reduction of the population mean firing rate to 25 ± 4% (1 µM baclofen, see raster plots in **Figure 1—figure supplement 1D**) and to 1.2 ± 0.4% (10 µM baclofen) of baseline values (**Figure 1H**). Thus, we can use this system to study how chronic changes in the GABA$_B$R-mediated neuromodulation impact properties of individual synapses, single neurons, and neural networks.

## Stability of firing rates at the population level

To assess how GABA$_B$R activation affects firing properties of the population at long timescales, we measured spiking activity during a baseline recording period and for 2 days following the application of 10 µM baclofen. As expected, baclofen caused a fast and pronounced drop in firing rates to 1.2 ± 0.4% of baseline values (**Figure 2A,C,D**). We hypothesized that if the network has an intrinsically regulated firing rate 'set point', as suggested by theories of homeostatic plasticity, we would see an increase in firing rates back to baseline values after administration of baclofen. Indeed, firing rates gradually returned to baseline values, reaching 57 ± 8% and 103 ± 14% of baseline after 1 and 2 days, respectively (**Figure 2C,D**). Notably, the average firing rates of networks under control conditions showed changes of similar magnitude during two recording days (without baclofen application, 113 ± 9% of baseline after 2 days; **Figure 2B–D**). Moreover, the characteristic log-normal distributions of single-unit firing patterns were essentially identical before and 2 days after baclofen application (**Figure 2E**, p = 0.66; Kolmogorov–Smirnov test). Importantly, baclofen remains potent after over 2 days of exposure to recording conditions (**Figure 2—figure supplement 1A**). Additionally, washout of baclofen following 2 days caused a significant increase in firing rate, indicating sustained activity of both baclofen and the GABA$_B$R-induced G-protein-mediated signaling (**Figure 2—figure supplement 1B–D**). Moreover, the magnitude of the GABA$_B$R-mediated presynaptic inhibition, as measured by FM dyes, was not reduced following washout (**Figure 2—figure supplement 1E,F**), indicating that GABA$_B$Rs did not undergo desensitization under our experimental conditions. Taken together, these results demonstrate a robust and precise compensatory response at the level of population averages, confirming the idea that neuronal networks grown in vitro (**Turrigiano et al., 1998**), similar to the circuits in vivo (**Hengen et al., 2013**; **Keck et al., 2013**), have the ability to homeostatically regulate their mean firing rate.

## Single units are not stable

After establishing that mean firing rates of hippocampal networks are precisely restored even after a pronounced, ongoing perturbation, we asked whether this stability is maintained at the level of the individual neurons that comprise the network. First, we analyzed firing rates of individual units during

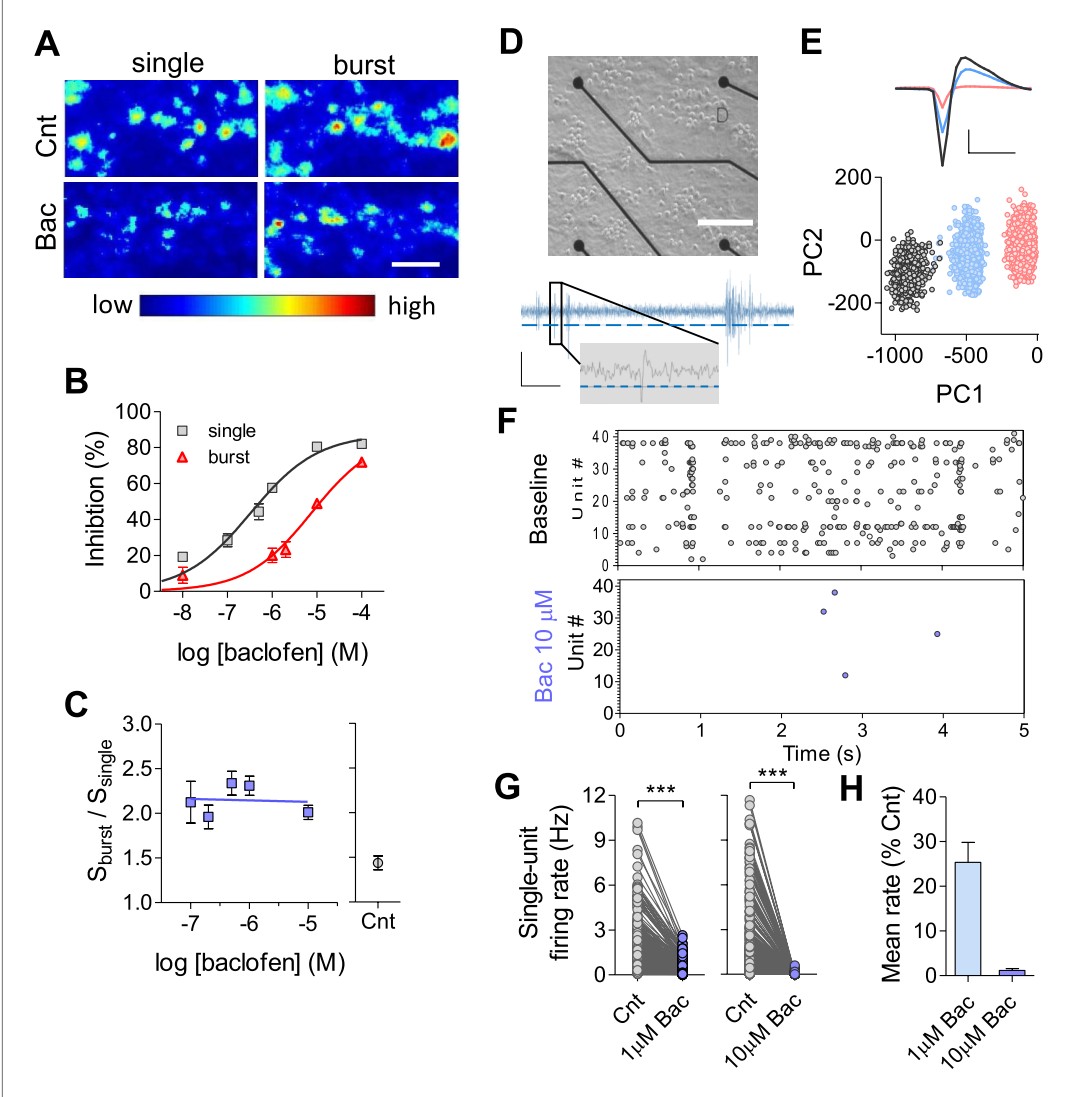

**Figure 1**. Quantifying GABA$_B$R-mediated synaptic inhibition at synaptic, single neuron and network levels. (**A**) Representative Δ*F* images obtained by single (30 APs@0.2 Hz) and burst (30 APs@6 bursts, each burst contained 5 APs; inter-spike-interval, 10 ms; inter-burst-interval, 5 s) stimulations before and 10 min after 10 μM baclofen application. Scale bar: 5 μm. (**B**) Dose–response curve for the inhibitory presynaptic effect of baclofen during single vs burst stimulation patterns. Note, shift in the apparent IC$_{50}$ from 0.33 μM in single stimulation (n = 7–12) to 7.6 μM for burst stimulation type (n = 8–10). (**C**) Short-term plasticity detected by FM method (calculated as S$_{burst}$/S$_{single}$) was in the range of 2.0 ± 0.01 to 2.3 ± 0.01 for baclofen concentrations in the range of 0.1–10 μM (n = 10–14) comparing to plasticity of control (1.4 ± 0.07, n = 13). (**D**) *Top*: image of dissociated hippocampal culture plated on MEA. Black circles at the end of the black lines are the recording electrodes. Scale bar: 200 μm. *Bottom*: representative trace of recording from one MEA channel. Spikes are detected based on a set threshold (blue dashed line). *Inset* is a zoom to one detected spike. Scale bars: 8 μV, 35 ms (6 ms for insert). (**E**) Example of spike sorting for one channel. *Bottom*: each waveform is represented in principal component space forming three distinct clusters. *Top*: mean waveforms for each cluster. Scale bar: 10 μV, 1 ms. (**F**) Representative raster plots of MEA recording before and immediately after application of 10 μM baclofen (***p < 0.0001, unpaired, two-tailed Student's *t*-test). (**G**) Acute effect of baclofen on firing rate at the single-unit level. *Left*: 1 μM baclofen (n = 5, 260 units). *Right*: 10 μM baclofen (n = 5, 314 units). (**H**) Acute effect of baclofen on the mean firing rate of the population (the same data as in **G**). Error bars represent SEM.

The following figure supplement is available for figure 1:

**Figure supplement 1**. Properties of baclofen-induced changes in synaptic dynamics and single-unit firing in hippocampal neurons.

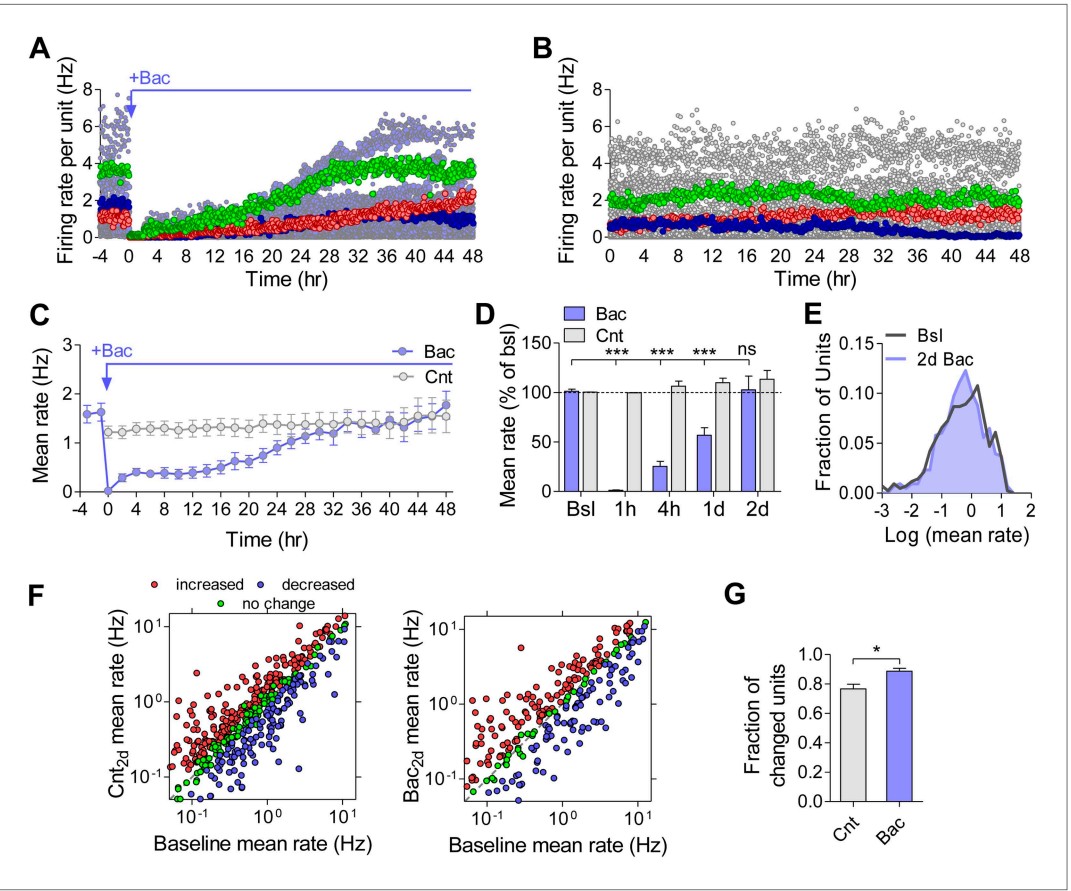

**Figure 2**. Homeostatic regulation of firing rates is more precise at the network, than the single-unit level. (**A**) Analysis of the firing rate of each unit in a representative MEA experiment over the course of 2 days of recording in the presence of 10 μM baclofen. Representative units that precisely returned (green), increased (red), and decreased (blue) relative to baseline are highlighted. (**B**) Analysis of the firing rate of each unit in a representative MEA experiment over the course of 2 days of recording under control conditions. Representative units that didn't change (green), increased (red), and decreased (blue) relative to baseline are highlighted. (**C**) Mean firing rate of 48 hr control (grey, n = 7) and baclofen (blue, n = 5) MEA recordings. 3 hr of baseline rate are shown for baclofen experiments. (For clarity, only every other hour is shown.) Error bars represent SEM. (**D**) Statistical comparison of the representative time points (the same data as in **C**). Error bars represent SEM. (***$p < 0.0001$, baclofen compared to baseline; all control hours were not significantly changed; repeated-measures ANOVA with Bonferroni's multiple comparison test.) (**E**) Distribution of unit firing rates (log scale) during baseline and after 2 days in the presence of baclofen. (**F**) Per unit correlation between baseline firing rates and firing rates after 2 days: *Left*: control (n = 7; 490 units); *Right*: in the presence of baclofen (n = 5; 314 units). Colors represent units that significantly increased (red), decreased (blue), or remained stable (green) as determined by bootstrapping (see 'Materials and methods' for details). Note log scale of both axes. (**G**) Summary of data in F (*$p < 0.05$; unpaired, two-tailed Student's *t*-test). Error bars represent SEM.

The following figure supplements are available for figure 2:

**Figure supplement 1**. Prolonged exposure to baclofen does not cause reduction in the sensitivity of synapses and neurons to the GABA$_B$R-mediated inhibition.

**Figure supplement 2**. Firing rate homeostasis is not precise at the level of multi-units.

**Figure supplement 3**. Effect of GABA uptake inhibitor on mean firing rate.

**Figure supplement 4**. MEA analyses are robust over different parameters.

2 days of recording under control condition. While a high correlation was observed between firing rates at the baseline and after 2 days (Spearman r = 0.8, p < 0.0001, n = 490), a large fraction of units did not return to their baseline values (*Figure 2F*, left). Only 23 ± 3% of units remained significantly unchanged during 2 days of control recordings, while 77 ± 3% changed their firing rates significantly (*Figure 2G*). This phenomenon became even more pronounced after the application of baclofen, with only 11 ± 2% of units returning to the baseline, while 89 ± 2% were significantly changed (*Figure 2F* right and 2G). The change in firing rate after 2 days of recordings was negatively correlated with the initial firing rate of the same unit under both control (*Figure 1—figure supplement 1E*; Spearman r = −0.29, p < 0.0001, n = 490) and baclofen (*Figure 1—figure supplement 1F*; Spearman r = −0.47, p < 0.0001, n = 305) conditions, due to a larger relative variability of low firing units. Notably, when analyzing multiunit recordings per electrode, firing rates after 2 days of recording displayed statistically significant change from the baseline in ~80% of the cases (*Figure 2—figure supplement 2A,B*). To ensure that per unit variability is not due to a possible movement of neurons in and out from recording electrodes, we measured the movement of individual neurons over the course of 2 days and found that the mean movement of neurons (1.6 ± 0.1 µM) was negligible in comparison to the size of the electrode (30 µM) and neuron's cell body (11.4 ± 0.3 µM). These results confirm that the observed variability at the level of single units does not result from imperfect spike sorting or neuronal mobility.

Taken together, these data indicate the existence of significant dynamics at the level of the individual hippocampal neurons in mature hippocampal cultures, operating under the constraint that the overall firing rates do not change. The observed instability of firing rates at the individual neuron level becomes even more pronounced following the perturbation. Thus, the same mean firing rate of the population arises from variable firing rates of individual neurons.

## Calcium imaging confirms the variability of individual firing rates

As fluorescent calcium sensors are widely used to image neural activity, we performed somatic calcium imaging in visually identified individual neurons to verify whether the stabilization of population mean firing rates emerges from variable firing rates of single neurons following chronic perturbation. We utilized the genetically encoded $Ca^{2+}$ indicator GCaMP6f that displays single-action-potential sensitivity and broad dynamic range (*Chen et al., 2013*). An AAV-based gene delivery system under the synapsin promoter was used to express GCaMP6f in hippocampal neurons. GCaMP6f-expressing neurons displayed spontaneous $Ca^{2+}$ dynamics typically consisting of transients of varying amplitudes corresponding to single spikes and spike bursts (*Figure 3A*). We first established the relationship between the calcium transients and spiking activity. Single spikes induced robust calcium transients (24.6 ± 1.4% $\Delta F/F$; *Figure 3—figure supplement 1A*). Although the exact number of spikes could not be accurately predicted from the size of the calcium transients, high-frequency bursts consisting of 2–10 spikes at 100 Hz induced significantly higher $\Delta F/F$ peak amplitudes as compared with single spikes and the average $\Delta F/F$ linearly correlated with the number of spikes (*Figure 3—figure supplement 1B,C*). Thus, the $\Delta F/F$ peak amplitude of calcium transients is approximately proportional to the number of spikes that triggered it.

To examine the stability of single neurons and neuronal populations at extended timescales, we performed time-lapse imaging of somatic $Ca^{2+}$ dynamics during a baseline recording period, after acute baclofen application and for 2 days following baclofen application. Each imaging session had a duration of 20 min, fairly representing firing stability of several hours (*Figure 2—figure supplement 4*). Analysis of $Ca^{2+}$ dynamics in 192 neurons shows that 10 µM baclofen blocked $Ca^{2+}$ transients acutely, while 48 hr after baclofen incubation the mean $Ca^{2+}$-transient amplitude returned to the baseline level (*Figure 3A,C*). Notably, the distributions of single-neuron $Ca^{2+}$-transient sizes were skewed towards low-activity levels (*Figure 3D*) as seen for electrophysiological measurements of spiking activity, indicating that the population of visually selected neurons does not represent a frequency-dependent bias. In addition, the distributions were indistinguishable before and 2 days after baclofen application (*Figure 3D*). Despite stability of $Ca^{2+}$ dynamics at the population level, 54 ± 6% of neurons did not return to their baseline values 2 days after baclofen application (*Figure 3E,F*). The change in $Ca^{2+}$-transient amplitudes after 2 days of baclofen incubation was negatively correlated with the initial amplitudes at the same neuron (*Figure 3—figure supplement 1F*). Thus, calcium imaging in identified neurons confirms the electrophysiological data, strengthening our conclusion regarding the stabilization of population dynamics despite imprecise homeostatic compensations at the level of single neurons.

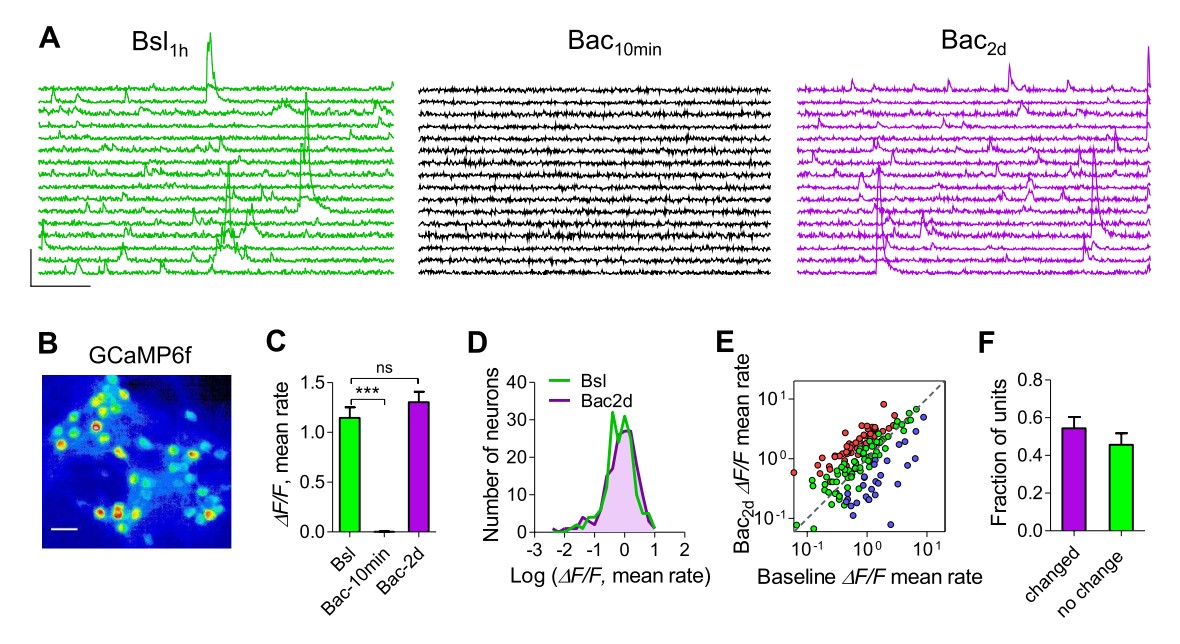

**Figure 3**. Calcium imaging confirms more precise homeostatic regulation of firing rates at the network, than the single-unit level. (**A**) Representative traces ($\Delta F/F$) showing the same 16 neurons before, 10 min after 10 µM baclofen application, and after 2 days in the presence of baclofen. Bar scales: 50% $\Delta F/F$, 5 s. (**B**) Pseudo-color coded image showing cultured neurons expressing GCaMP6f. Scale bar: 50 µm. (**C**) Summary of the mean rate (averaged peak amplitude per min) before (green), after 10 min (black) and following 2 days (purple) of baclofen application. Population mean rates were restored after 2 days (n = 4, 192 neurons). (**D**) Distribution of neuron $\Delta F/F$ rates (log scale) during baseline and after 2 days in the presence of baclofen. (**E**) Per neuron correlation between baseline firing rates and firing rates after 2 days in the presence of baclofen (n = 4; 192 neurons). Colors represent neurons that significantly increased (red), decreased (blue), or remained stable (green) as determined by bootstrapping. Note log scale of both axes. (**F**) Summary of data in **E** (54.3% ± 6 cells were significantly changed). Error bars represent SEM.

The following figure supplement is available for figure 3:

**Figure supplement 1**. Calcium imaging using GCaMP6f sensor in cultured hippocampal neurons.

## Population firing synchrony is stable despite variability of single-unit patterns

We next investigated the persistence of temporal firing patterns in hippocampal networks. We utilized electrophysiological recordings by MEAs that enable superior temporal resolution compared to calcium imaging. *Figure 4A* presents raster plots during periods of baseline, 4 hr and 2 days following baclofen application in a single experiment. We first ensured that there was no significant change in firing synchrony, as estimated by the fraction of spikes participating in network-bursts and burst duration (*Figure 4B*, see 'Materials and methods' for burst detection), under control conditions. Baclofen transiently increased population firing synchrony by increasing the fraction of spikes participating in bursts, burst duration (*Figure 4B,C*), and number of spikes per network burst (*Figure 4—figure supplement 1A*). The observed GABA$_B$R-mediated increase in firing synchrony returned to baseline values after 14 hr (*Figure 4B,C*). These results indicate that the population-burst pattern, similar to the population firing rate, undergoes a precise homeostatic compensation.

We then asked whether the observed stability of population firing synchrony results from stability of spike patterns at the single-unit level. Here too, we ensured that there was no change in single-unit bursts under control conditions (*Figure 4D,E* and *Figure 4—figure supplement 1B*). Interestingly, following baclofen application, the average fraction of spikes participating in single-unit bursts returned to baseline levels with dynamics very similar to those of network bursts (*Figure 4D*). However, single-unit burst analysis per individual unit shows a high degree of variability. Under control conditions, 42 ± 2% of units significantly changed the fraction of spikes in bursts, while 56 ± 2% remained unchanged. 2 days after baclofen application, 63 ± 8% were significantly changed, while only 37 ± 8%

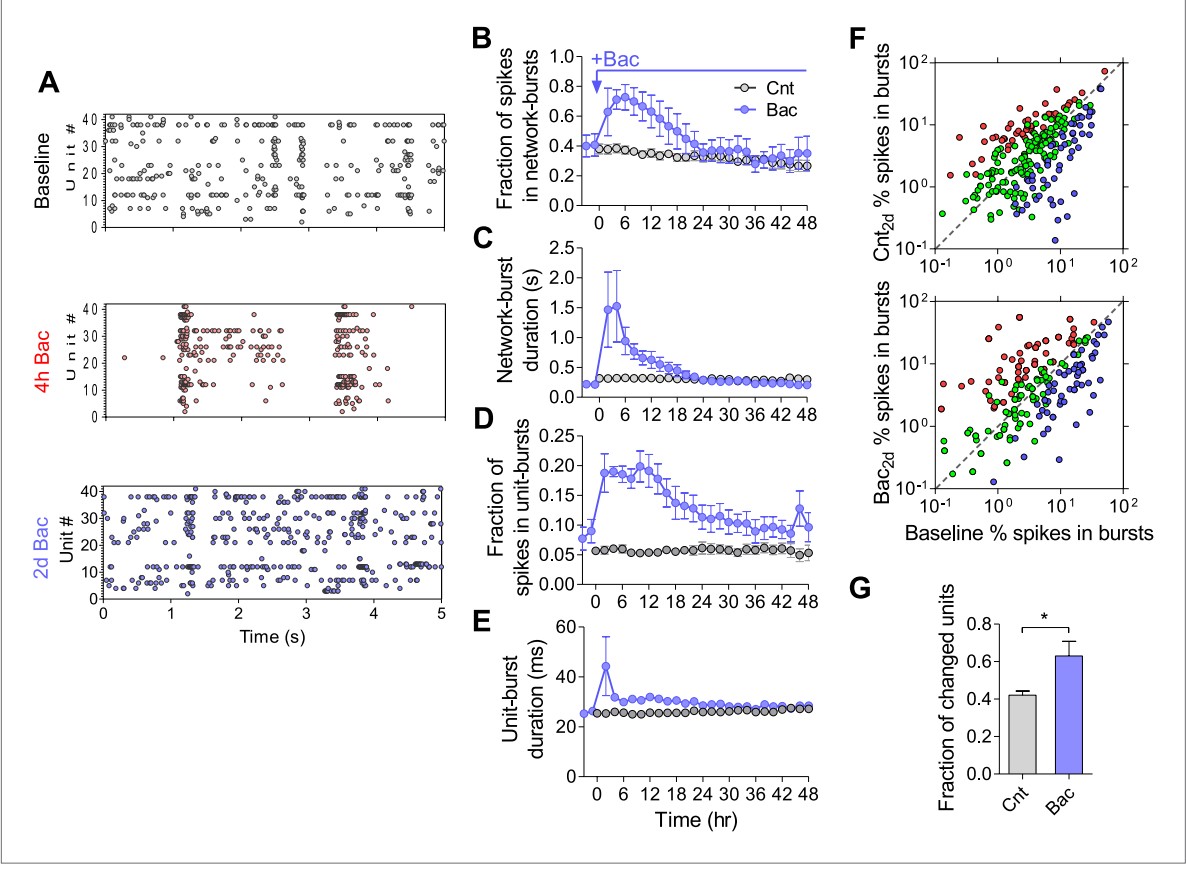

**Figure 4**. Temporal firing patterns are homeostatically regulated. (**A**) Representative raster plot of MEA recording before, 4 hr and 2 days after application of baclofen. (**B**) Baclofen causes a transient increase in fraction of spikes participating in network bursts (hours 2–4, p < 0.01; hours 4–10, p < 0.001; hour 12, p < 0.01; hour 14, p < 0.05; repeated-measures ANOVA with Bonferroni's multiple comparison test, compared to baseline values). The number of spikes that are part of network bursts was divided by the total number of spikes. (**C**) Baclofen causes a transient, short-lived, increase in duration of network bursts (hours 2–4, p < 0.001; hour 6, p < 0.05; repeated-measures ANOVA with Bonferroni's multiple comparison test, compared to baseline values). (**D**) Baclofen causes a transient increase in fraction of spikes participating in single-unit bursts (hours 2–4, p < 0.01; hours 4–12, p < 0.001; hour 14, p < 0.01; repeated-measures ANOVA with Bonferroni's multiple comparison test, compared to baseline values). The number of spikes that are part of single-unit bursts was divided by the total number of spikes of that unit. (**E**) Baclofen causes a transient, short-lived, increase in duration of single-unit bursts (hours 2, p < 0.001; repeated-measures ANOVA with Bonferroni's multiple comparison test, compared to baseline values). (**F**) Per unit correlation of fraction of spikes in single-unit bursts between baseline and 2 days after baclofen application: *Top*: control (n = 7; 279 units); *Bottom*: in the presence of baclofen (n = 5; 234 units). Colors represent units that significantly increased (red), decreased (blue), or remained stable (green) as determined by bootstrapping (see 'Materials and methods' for details). Note log scale of both axes. (**G**) Summary of data in (**D**) (*p < 0.05, unpaired, two-tailed Student's *t*-test). Error bars represent SEM.

The following figure supplements are available for figure 4:

**Figure supplement 1**. Number of spikes in network- and single-unit bursts in the presence of baclofen.

**Figure supplement 2**. Single-unit burst characteristics are stable across a large range of thresholds.

of units remained unchanged (***Figure 4F,G***). Notably, bursts characteristics were not significantly changed, outside of the first 2 hr following baclofen application (***Figure 4E*** and ***Figure 4—figure supplement 1B***). These data indicate that firing synchrony of spontaneous activity is maintained at the network level, despite large changes in burst patterns at the single-unit level.

## Changes in the inhibition–excitation ratio occur in parallel to firing rate fluctuations

The inhibition–excitation (I/E) ratio constitutes an important factor in firing rate homeostasis (***Liu, 2004***; ***Maffei et al., 2004***; ***Maffei and Turrigiano, 2008***). To assess the effect of chronic increase in the GABA$_B$R

activity on the I/E ratio, we isolated the spontaneous excitatory and inhibitory currents (sEPSCs, sIPSCs, respectively) at the same cell based on the reversal potentials of AMPAR-mediated excitatory and GABA$_A$R-mediated inhibitory currents (*Figure 5A,B*). We then calculated the integrated excitatory and inhibitory conductances (G$_E$, G$_I$, respectively; see 'Materials and methods').

As expected, acute application of baclofen almost completely blocked both sEPSCs and sIPSCs, from 0.40 ± 0.07 and 0.87 ± 0.1 nS to 0.02 ± 0.006 and 0.06 ± 0.02 nS for G$_E$ and G$_I$, respectively (*Figure 5C,D*). After 4 hr of incubation with baclofen, G$_E$ and G$_I$ partially recovered to 0.16 ± 0.07 nS and 0.61 ± 0.2 nS, respectively (*Figure 5C,D*), suggesting a faster rate of G$_I$ homeostatic regulation. Finally, after 2 days in the presence of baclofen, the network showed full recovery with G$_E$ reaching 0.44 ± 0.1 nS and G$_I$ reaching 1.02 ± 0.2 nS ($p > 0.8$, *Figure 5C,D*). As a result, I/E ratio was transiently increased from 3.1 ± 0.4 to 9.9 ± 2.8 following 4 hr baclofen incubation ($p < 0.001$), returning to the baseline levels following 2 days of baclofen (3.04 ± 0.5, $p > 0.5$; *Figure 5E*). These data show that I/E balance of the network is tightly regulated, supporting homeostatic restoration of spontaneous spiking activity of the network even under a constant increase in the GABA$_B$R activity.

## Synaptic homeostatic mechanisms: increase in mEPSC frequency and amplitude

Having observed a recovery of the firing properties at the population level, we next investigated which homeostatic mechanisms were underlying these changes. To accomplish this, we recorded mEPSCs from hippocampal neurons under control conditions and at different time points following baclofen application (*Figure 6A*). While acute baclofen application did not have a significant effect on mEPSCs (*Figure 6—figure supplement 1*) as has been reported previously (*Lei and McBain, 2003*), chronic baclofen incubation triggered gradual changes in mEPSC frequency and amplitude. We did not observe a significant increase in mEPSC amplitude 4 hr after baclofen application (24.7 ± 1.3 and 27.7 ± 3.3 pA in control and 4 hr after baclofen, respectively, $p = 0.46$). However, a 1.25-fold increase to 30.8 ± 2.2 pA ($p < 0.05$) was detected 2 days following baclofen application (*Figure 6B,C*). This was coupled with two-fold increase in the frequency of mEPSCs from 2.1 ± 0.7 to 4.3 ± 1.5 Hz ($p < 0.05$) following 4 hr baclofen application and a large 4.8-fold increase to 10 ± 3.2 Hz ($p < 0.01$) following 2 days (*Figure 6D,E*). These data indicate that both pre- and postsynaptic modifications of quantal excitatory synaptic transmission occur following chronic, use-dependent inhibition of the evoked synaptic activity.

## Increase in intrinsic excitability in response to chronic activity suppression

In addition to synaptic homeostatic mechanisms, modulation of intrinsic excitability is an important facet of neuronal adaptation (*Desai et al., 1999*; *Kim and Tsien, 2008*; *Maffei and Turrigiano, 2008*). To examine the effect of chronic GABA$_B$R activation on intrinsic electrophysiological properties of hippocampal neurons, we incubated cultures with 10 µM baclofen for 4 hr and 2 days. We then elicited action potentials (APs) in response to increasing somatic current injections ranging from −40 to +180 pA (F–I curves) in the presence of postsynaptic receptor blockers. While there was no difference in F–I curves between control and 4 hr baclofen incubation, 2 days of baclofen incubation caused a significant leftward shift of the curve (*Figure 7A,B*). Additionally, resting membrane potential (RMP, *Figure 7C*) was −64.8 ± 1.7 mV in untreated neurons, became more depolarized already after 4 hr (−58.2 ± 2.2, $p < 0.05$) and further depolarized after 2 days (−55.4 ± 1.5 mV, $p < 0.001$). Finally, input resistance (R$_{in}$), determined by the voltage response to increasing somatic current injections ranging from −80 to −20 pA, showed a tendency towards larger values after 4 hr and was significantly increased following 2 days baclofen incubation (from 233 ± 14 to 373 ± 59 MΩ, $p < 0.01$; *Figure 7D,E*). These data show that, in addition to synaptic modifications, increased intrinsic excitability contributes to the homeostatic restoration of network firing properties following chronic GABA$_B$R-mediated inhibition.

## Short-term synaptic plasticity is not homeostatically restored

To assess whether the increase in mEPSC frequency is paralleled by modifications in the evoked synapse release probability, we quantified basal synaptic vesicle release at different time points following baclofen application utilizing FM1-43 dye (*Abramov et al., 2009*). To this end, we quantified the total amount of releasable fluorescence at each bouton (Δ$F$) and the density of FM-positive puncta per image ($D$) following stimulation (30 stimuli at 1 Hz) in the presence of 10 µM FM1-43. Our results demonstrate a 1.5- and 1.8-fold increase in Δ$F$ across synaptic populations following 4 hr and 2 days, respectively ($p < 0.001$, *Figure 8A,B*). To confirm that prolonged incubation with baclofen increases

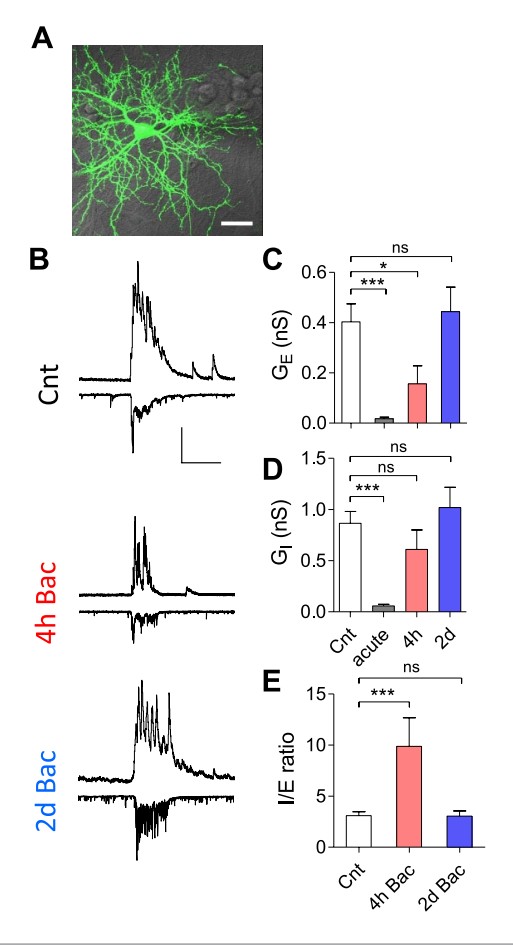

**Figure 5**. Dynamics of I/E ratio following chronic GABA<sub>B</sub>R-mediated inhibition. (**A**) Image of patched neuron. Alexa-fluor 488 (10 µM) was added to the patch pipette for imaging. Scale bar: 20 µm. (**B**) Representative traces of sEPSCs (−65 mV holding potential, *bottom*) and sIPSCs (+10 mV holding potential, *top*) for control, 4 hr and 2 days preincubation with baclofen. Measurements of sEPSCs following baclofen pre-incubation were done in the presence of baclofen. (**C**) Mean integrated excitatory conductance ($G_E$) in control (n = 32), following acute (n = 10), 4 hr (n = 11) and 2 days (n = 16) baclofen application. Excitatory conductance is completely restored following 2 days of exposure to baclofen. Error bars represent SEM. (*p < 0.05, ***p < 0.0001; Kruskal–Wallis test with Dunn's multiple comparison test). (**D**) Mean integrated inhibitory conductance ($G_I$, same cells as **B**). Inhibitory conductance is completely restored as well following 2 days of exposure to baclofen. Error bars represent SEM. (***p < 0.001; Kruskal–Wallis test with Dunn's multiple comparison test). (**E**) Mean I/E ratio per neuron (same cells as in **C**, **D**). The I/E ratio of each cell was calculated and the resulting ratios were averaged. *p < 0.05, **p < 0.01, ***p < 0.001; one-way ANOVA with Dunnett's multiple comparison test. Error bars represent SEM.

synaptic vesicle exocytosis, the total pool of recycling vesicles was stained by maximal stimulation (600 stimuli at 10 Hz) and subsequently destained by 1 Hz stimulation. The destaining rate constant (measured as $1/\tau_{decay}$, where $\tau_{decay}$ is an exponential time course) increased by ~36% following 2 days of baclofen incubation (p < 0.01; *Figure 8—figure supplement 1*). Thus, an increase in the release probability of hippocampal boutons constitutes an adaptive mechanism stabilizing firing properties of hippocampal network.

Given an inverse correlation between release probability and short-term synaptic facilitation (*Debanne et al., 1996*; *Dobrunz and Stevens, 1997*), homeostasis of mean firing rate is expected to be paralleled by concurrent changes in short-term synaptic plasticity. The total presynaptic strength within a given region of the hippocampal network (*S*) can be estimated as the product of Δ*F* and *D* (*S* = Δ*F* × *D*). The magnitude and the sign of short-term presynaptic plasticity ($S_{burst}/S_{single}$) were calculated by dividing the total number of vesicles recycled due to bursts by the number of vesicles recycled by a similar number of single spikes in the same population of synapses (*Figure 8C*). Indeed, short-term synaptic facilitation was decreased from 1.9- to 1.2-fold after 4 hr (p < 0.001, *Figure 8D,E*) and was completely abolished following 2 days (p < 0.001, *Figure 8D,E*) of incubation with baclofen. These results demonstrate that short-term synaptic plasticity is not preserved under homeostasis of population firing properties.

## Uncompensated reduction in firing rate and synchrony by AMPAR blockade

Having established a relationship between the use-dependent blockade of synaptic transmission via GABA<sub>B</sub>Rs and spontaneous firing properties, we asked whether use-independent blockade of synaptic transmission triggers similar effects. For this purpose, we blocked fast excitatory synaptic transmission by using the AMPA receptor (AMPAR) blocker CNQX. While previous studies demonstrated that chronic AMPAR blockade increases both the frequency and the amplitude of mEPSCs (*Thiagarajan et al., 2002*) already 4 hr after application of the antagonist (*Jakawich et al., 2010*), its effect on firing properties of the network has not been explored yet.

CNQX (10 µM) caused a reduction in the population mean firing rate to 52 ± 12% of baseline values immediately after application (p < 0.01; *Figure 9B*). Application of NMDAR and AMPAR blockers together resulted in ~80% of firing rate inhibition (*Figure 9—figure supplement 1*),

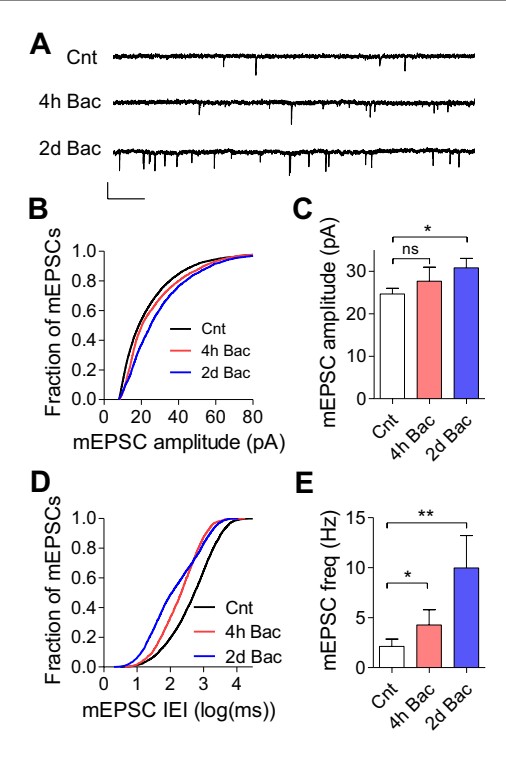

**Figure 6**. Chronic GABA_BR-mediated inhibition triggers an increase in mEPSC frequency and amplitude. (**A**) Representative traces of mEPSCs for control, 4 hr and 2 day incubations in baclofen. Scale bar: 40 pA, 200 ms. Measurements of mEPSCs were done in the presence of baclofen. (**B**) Cumulative histograms of mEPSC amplitudes in control (n = 30) and following 4 hr (n = 11) and 2 days (n = 15) of incubation with baclofen. The mean of mEPSC amplitude increased from 25.4 pA in control to 27.7 and 30.8 pA following 4 hr and 2 days of baclofen application, respectively. (**C**) Summary of data in **B**. Mean mEPSC amplitude is significantly elevated 1.25-fold (p < 0.05) only after 2 days in baclofen. Error bars represent SEM. *p < 0.05; one-way ANOVA with Dunnett's multiple comparison test. (**D**) Cumulative histogram of mEPSC inter-event intervals showing a gradual shift to smaller values from control through 4 hr baclofen to 2 days baclofen incubation (the same experiments as in **C**). (**E**) Summary of data in **D**. mEPSC frequency is increased twofold after 4 hr (p < 0.05) and 4.6-fold after 2 days (p < 0.01) incubation in baclofen. *p < 0.05, **p < 0.01; one-way ANOVA with Dunnett's multiple comparison test. Error bars represent SEM.

The following figure supplement is available for figure 6:

**Figure supplement 1**. Baclofen does not affect mEPSC frequency and amplitude acutely.

indicating that NMDARs contribute ~30% to spontaneous spiking activity. The residual ~20% may arise from electrical coupling via gap junctions (*Hormuzdi et al., 2001*), intrinsically bursting neurons (*Yue and Yaari, 2004*), ephaptic effects, and/or rebound spiking following inhibition. While CNQX triggered a compensatory increase in synapse release probability already 4 hr after its application (*Figure 9F,G* and (*Jakawich et al., 2010*)), the population mean firing rate remained largely uncompensated, staying at 40 ± 5% of baseline 2 days following the perturbation (*Figure 9A,B*). Likewise, there was a significant shift in the single-unit firing rates towards lower frequencies after 2 days (p < 0.001, Kolmogorov–Smirnov test; *Figure 9C*). When examining changes at the individual unit level, we found that only 16 ± 8% of units remained significantly unchanged during 2 days of the perturbation, while 71 ± 8% decreased their firing rates (p < 0.001, *Figure 9D,E*). It is noteworthy that CNQX and AMPAR-mediated signaling remain active even after 2 days incubation as evidenced by the increase in network and single unit firing rate following washout (*Figure 9—figure supplement 2*).

Analysis of firing pattern shows that CNQX caused a permanent reduction in the fraction of spikes participating in population bursts (*Figure 9A,H*), as well as in the number of spikes comprising population bursts (*Figure 9—figure supplement 3A*), reflecting a reduction in firing synchrony. Population-burst duration, however, showed a brief increase followed by a gradual, non-significant, decrease (*Figure 9—figure supplement 3B*). A similar effect was observed at the single-unit level regarding the fraction of spikes participating in single-unit bursts, while burst properties were not significantly affected (*Figure 9I* and *Figure 9—figure supplement 3C,D*). Taken together, these results suggest that use-independent postsynaptic blockade of excitatory synaptic transmission induces firing rate reduction with concomitant desynchronization of population firing. Although blockade of AMPAR-mediated excitatory synaptic transmission had a lower impact on firing rates than frequency-dependent synaptic inhibition via GABA_BRs (*Figure 2E*), this effect cannot be efficiently compensated in the network.

## Discussion

How neural circuits maintain the balance between stability and plasticity is one of the most intriguing questions in neuroscience. Taking into account the highly dynamic and heterogeneous nature of individual synapses, ion channels, transmitters,

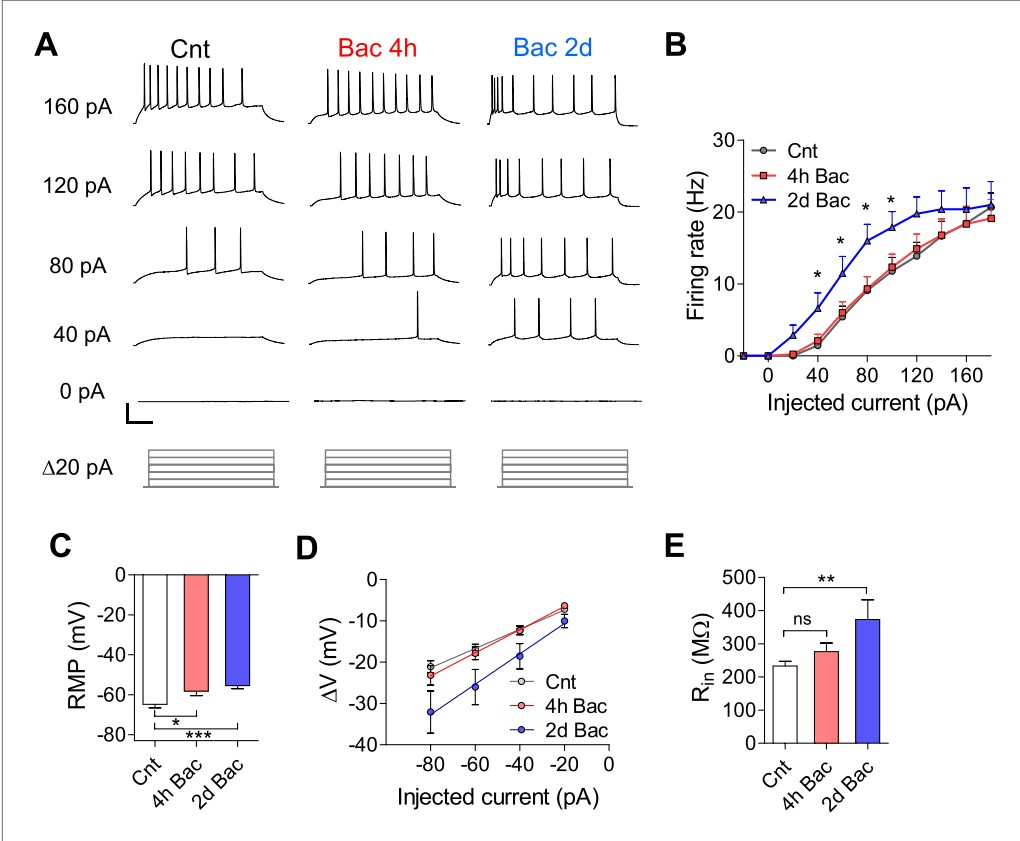

**Figure 7**. Intrinsic excitability is increased in response to activity suppression. (**A**) Representative traces of voltage responses evoked by 20 pA step of current injections after control, 4 hr and 2 days baclofen incubation, elicited from RMP (scale bars: 40 mV, 100 ms). (**B**) F–I relationship after control, 4 hr and 2 days baclofen incubation. After 2 days incubation there is a significant leftward shift of the curve showing greater excitability (control, n = 18; 4 hr, n = 18; 2 days, n = 16; *p < 0.05) following long-term GABA$_B$R activation. (**C**) RMP is depolarized after short baclofen incubation (control, n = 19; 4 hr, n = 21; 2 days, n = 14). (**D**) I–V curve (same cells as **C**). (**E**) R$_{in}$ is significantly increased following 2 day baclofen incubation (same cells as **C**).

and their receptors, understanding how individual neurons and neuronal populations maintain stable activity over long timescales or adjust their properties to changes in their external environment is an important and difficult challenge.

## Homeostatic control of single neurons and network functions

To relate the network's behavior to properties of its single components, we integrated, in this study, recordings with extracellular MEA, the intracellular patch-clamp recordings and high-resolution functional imaging at the single-synapse level. In particular, this system enables us to examine whether homeostatic mechanisms operating at a cell-autonomous level are sufficient to confer population firing stability (*Burrone et al., 2002*; *Goold and Nicoll, 2010*; *Turrigiano, 2012*) or whether homeostasis operates at the level of network-average properties.

Our results show that population spiking rates and patterns are intrinsically stable not only under basal conditions but also following profound activity perturbations. Despite a transient GABA$_B$R-mediated blockade of population firing rate with a simultaneous increase in firing synchrony, population activity was precisely restored to a 'set point' level after a period of 2 days. Interestingly, restoration of firing synchrony displays a faster kinetics than rebound of mean rates. In contrast to the observed firing macro-stability, single-unit behavior appears to be extremely dynamic. Only 23% of units displayed stable firing rates without perturbation, and this fraction was reduced to 11% following the perturbation. A complementary method, based on long-term recordings of somatic Ca$^{2+}$ dynamics, reflecting firing rates, in identified neurons showed that 46% of neurons returned to their baseline firing rates (*Figure 3*).

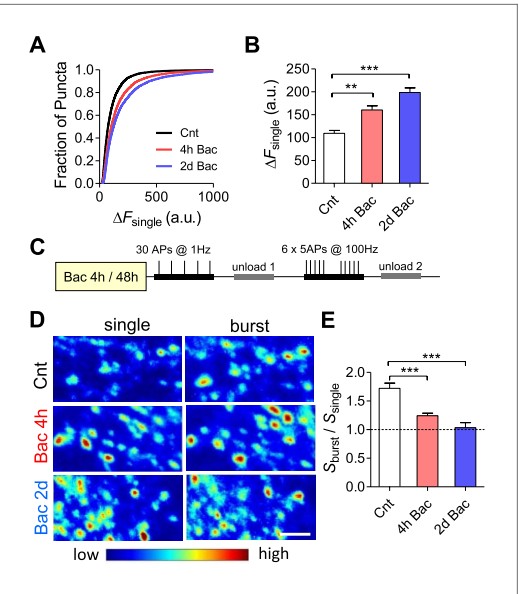

**Figure 8**. Short-term synaptic plasticity is not preserved in networks with similar firing properties. (**A**) Cumulative histogram of fluorescence intensity of FM stained puncta following 30 stimuli given at 1 Hz. ($\Delta F_{single}$ control, n = 15; 6099 puncta; 4 hr, n = 15, 6269 puncta; 2 days, n = 13; 5551 puncta.) (**B**) Summary of mean $\Delta F_{single}$ from (**A**). Mean $\Delta F_{single}$ is increased already after 4 hr of incubation with baclofen and remains high after 2 days of baclofen incubation. ***p < 0.001; one-way ANOVA with Tukey's multiple comparison test. (**C**) Experimental protocol used for STP experiments. (**D**) Representative images of FM1-43 staining from STP experiments. Note the increase in fluorescence intensity after 1 Hz stimulation following baclofen incubation. Scale bar: 5 µm. (**E**) The mean burst-to-single ratio of S is significantly decreased following baclofen incubation (control, n = 15; 4 hr, n = 15; 2 days, n = 13; p < 0.0001). ***p < 0.001; one-way ANOVA with Tukey's multiple comparison test. Error bars represent SEM.

The following figure supplement is available for figure 8:

**Figure supplement 1**. Synaptic vesicle exocytosis evoked by 1 Hz stimulation is increased after 2 days baclofen incubation.

Thus, two independent methodologies suggest that the majority of neurons display unstable firing rates even at extended timescales, confirming scale-invariant rate dynamics observed in a previous study (*Gal et al., 2010*). Thus, stability, as well as the compensatory feedback response, is greater at the population compared to the single-unit level. Moreover, quantal excitatory synaptic strength and intrinsic excitability were significantly affected by the perturbation and, thus, differ between the baseline and rebound periods. These results highlight the idea that similar network properties may arise from multiple configurations of individual components (*Prinz et al., 2004*; *Marder and Goaillard, 2006*). Most importantly, the observed differences in the precision of homeostatic regulation at different spatial scales suggest that population firing homeostasis is more than a sum of single-neuron adaptive responses, implying the existence of network-wide regulation rules (*Maffei and Fontanini, 2009*).

## The trigger of synaptic homeostatic responses

What is the trigger of compensatory synaptic responses? A variety of synaptic compensation mechanisms have been found following pharmacological or genetic perturbations in hippocampal and cortical neuronal cultures (*Turrigiano et al., 1998*; *Burrone et al., 2002*; *Thiagarajan et al., 2002*, *2005*; *Branco et al., 2008*; *Jakawich et al., 2010*), following sensory deprivation (*Maffei et al., 2004*; *Hengen et al., 2013*; *Keck et al., 2013*) or in a more physiological context of sleep (*Vyazovskiy et al., 2008*; *Lanté et al., 2011*). Interestingly, synaptic homeostasis hypothesis proposed by Tononi and colleagues (*Tononi and Cirelli, 2003*, *2014*) states that population firing synchrony, a hallmark of slow-wave sleep, contributes directly to homeostatic synaptic scaling. To examine if changes in population synchrony per se can trigger adaptive responses, we compared two perturbations producing similar (inhibitory) effects on the mean firing rate, while differentially affecting firing synchrony. For this purpose, we used baclofen that increases firing synchrony (*Figure 4*) and CNQX that induces firing desynchronization (*Figure 9H,I*). Both perturbations trigger the same types of synaptic adaptation at a comparable timescale: increase in the amplitude and the frequency of mEPSCs and in release probability (*Figures 6 and 9F,G* and (*Thiagarajan et al., 2002*; *Jakawich et al., 2010*)). Based on these results, we may conclude that (1) firing rates and patterns are independently regulated; (2) homeostatic systems generally sense a drop in spiking rates to induce an adaptive increase in excitatory synaptic transmission. Thus, our results in cultured neural networks don't support a causal relationship between firing synchrony and homeostatic synaptic response, suggesting that other factors, occurring during slow-wave sleep, may play a role in down-scaling of synapses in behaving animals.

Why network firing rates and patterns were not compensated following the AMPAR blockade requires future investigation. On the one hand, complete silencing of excitatory drive might exceed

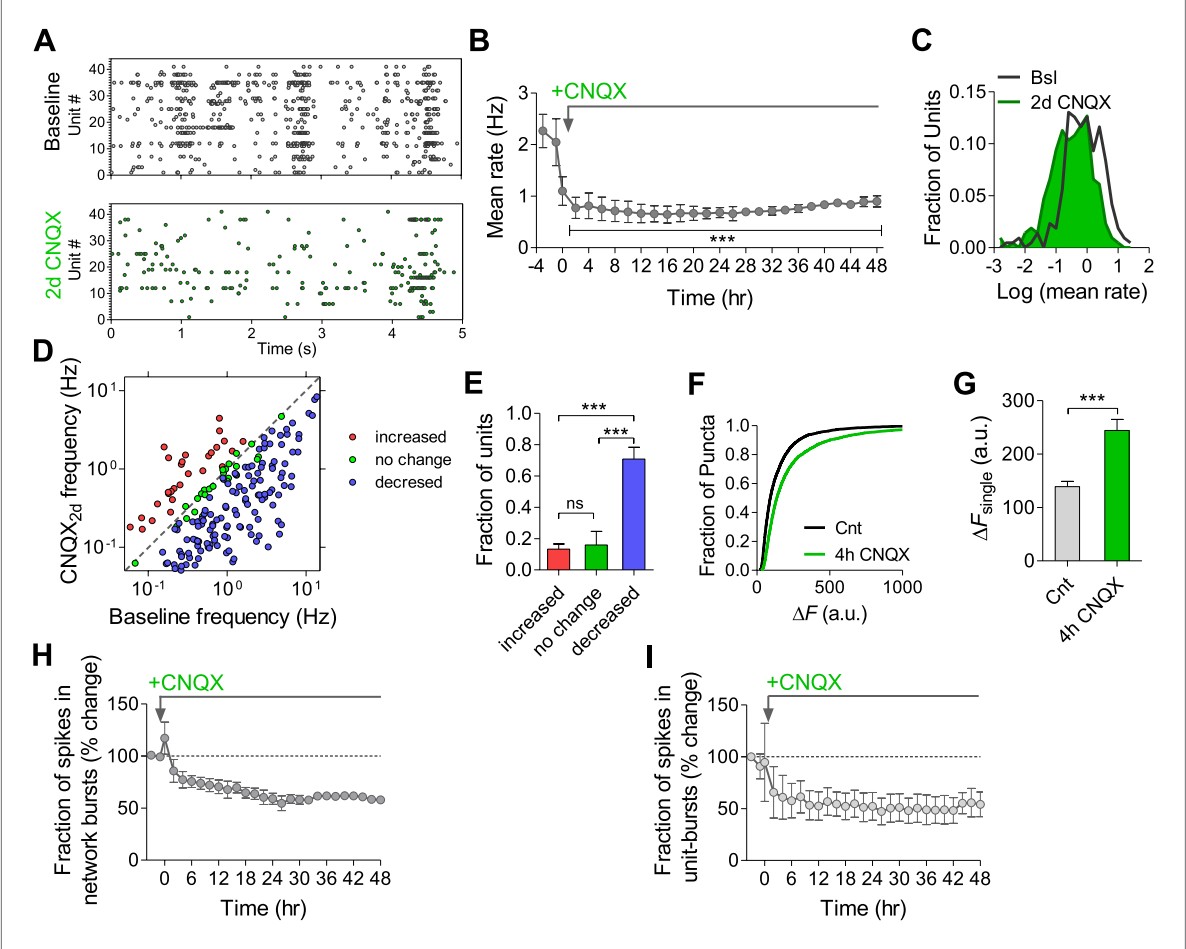

**Figure 9**. Effects of chronic AMPAR blockade on spontaneous network firing. (**A**) Representative raster plot of MEA recording before and 2 days after application of CNQX. (**B**) Changes in mean firing rate following 10 μM CNQX application utilizing MEA recordings (n = 4). 3 hr of baseline rates are shown. There is an immediate and prolonged reduction of firing rate (p < 0.001 for all hours compared to baseline; one-way ANOVA with Bonferroni's multiple comparison test). For clarity, only every other hour is shown. Error bars represent SEM. (**C**) Distribution of unit firing rates (log scale) during baseline and after 2 days in the presence of CNQX. (**D**) Per unit correlation between baseline firing rates and firing rates after 2 days in the presence of CNQX (n = 4, 128 units). Colors represent units that significantly increased (red), decreased (blue), or remained stable (green) as determined by bootstrapping (see 'Materials and methods' for details). Note log scale of both axes. (**E**) Summary of data in (**D**). ***p < 0.001; one-way ANOVA with Tukey's multiple comparison test. (**F**) Cumulative histogram of fluorescence intensity of FM1-43 stained puncta following 30 stimuli given at 1 Hz (ΔF$_{single}$ control, n = 9, 3631 puncta; 2 days CNQX, n = 9, 5251 puncta). (**G**) Summary of mean ΔF$_{single}$ from (**F**). Mean ΔF$_{single}$ is increased already after 4 hr of incubation with baclofen and remains high after 2 days of baclofen incubation. ***p < 0.001; one-way ANOVA with Tukey's multiple comparison test. (**H**) Normalized fraction of spikes in network bursts (the same experiments as in **B**). There was a significant reduction from the fourth hour onward (p < 0.001; repeated-measures ANOVA with Bonferroni's multiple comparison test). (**I**) Normalized fraction of spikes in single-unit bursts (the same experiments as in **B**). There was a significant reduction from the sixth hour onward (hours 6–10, p < 0.05; hours 12–36, p < 0.01; hours 36–48, p < 0.001; repeated-measures ANOVA with Bonferroni's multiple comparison test). Error bars represent SEM.

The following figure supplements are available for figure 9:

**Figure supplement 1**. Effect of AMPAR and NMDAR blockers on mean firing rate measured by MEA in hippocampal cultures.

**Figure supplement 2**. CNQX washout reveals an increase in the mean firing rate.

**Figure supplement 3**. Characteristics of network- and single-unit bursts in the presence of CNQX.

the capacity of the homeostatic system to compensate. On the other hand, the drop in the firing rate following baclofen application was even more pronounced; nevertheless, firing rate homeostasis was achieved. Similarly, blockade of GABA uptake via GAT-1 transporter resulted in a transient blockade of

firing rates that were precisely compensated 2 days following the perturbation (*Figure 2—figure supplement 3*). We can exclude the necessity of population bursts for induction of a proper profile of adaptive response since other treatments, such as ACh exposure, induce a reduction in synchrony that can be successfully compensated (*Kaufman et al., 2012*). Thus, our results indicate that adaptations at the level of intrinsic neuronal excitability and inhibitory drive are not sufficient to compensate the blockade of AMPAR-mediated excitatory drive. It is conceivable that AMPAR blockade might perturb the relationship between the firing rate and the intracellular $Ca^{2+}$ concentration, causing a failure in the regulatory system (*O'Leary et al., 2014*).

## Changes in neuronal properties do not always impact network performance

It is generally assumed that experience-dependent modifications of synaptic strength or intrinsic excitability are associated with functional changes in the network performance. Nevertheless, under some circumstances, synaptic and intrinsic modifications may have little effect on network functional properties (*Prinz et al., 2004*; *Thirumalai et al., 2006*). For example, networks utilizing uniform and multiplicative forms of postsynaptic scaling as a main compensatory response to the perturbation may preserve learning rules and information content transferred between neurons (*Turrigiano et al., 1998*; *Turrigiano and Nelson, 2004*).

While changes in neuromodulation via $GABA_BRs$ caused a profound increase in synapse release probability, intrinsic excitability, mEPSC frequency and amplitude, spontaneous spiking activity of the network returned to the baseline level during 2 days in the constant presence of the perturbation. However, the sensitivity of synaptic population to bursts was profoundly modified: while acute $GABA_BR$ activation by baclofen shifts synapses towards high-pass filters (*Figure 1A–C*), chronic baclofen application results in complete abolishment of short-term synaptic plasticity (*Figure 8D,E*), potentially reducing discrimination of input patterns by synaptic mechanisms. The reduction in the selectivity of synapses to afferent input may represent a trade-off between population firing stability and synaptic metaplasticity (*Abraham and Bear, 1996*; *Thiagarajan et al., 2007*). Thus, robust homeostatic control of ongoing population dynamics may coexist with unstable short-term synaptic plasticity.

In summary, our results suggest that invariant population mean rate and temporospatial coherence of spontaneous spiking can emerge from highly diverse combinations of synaptic strength and intrinsic neuronal properties. The observed micro-instability of individual neurons was truly intrinsic, taking place in a highly-controlled environment, irrespective of changes in experience, behavioral states, and interactions with higher-order supervising circuits. While firing macro-stability is robustly and accurately maintained by homeostatic control systems in the face of perturbations and uncertainties, the ability of synapses to discriminate input patterns was sacrificed. Thus, impairments of short-term synaptic plasticity and working memory functions, characterizing initial phases of numerous brain disorders, may be the tradeoff resulting from the system's efforts to maintain phenotypic stability of spontaneous population firing patterns. It remains to be seen whether hippocampal short-term synaptic plasticity can be maintained without sacrificing critical parameters of population dynamics.

## Materials and methods

### MEA preparation and recordings

Primary cultures of CA3–CA1 hippocampal neurons were prepared from newborn BALB/c mice on postnatal days 0–2, as described (*Slutsky et al., 2004*). The experiments were performed in 15–22 DIV cultures. All animal experiments were approved by the Tel Aviv University Committee on Animal Care. Cultures were plated on MEA plates containing 59 TiN recording and one internal reference electrodes (Multi Channel Systems [MCS], Germany). Electrodes are 30 μm in diameter and spaced 500 μm apart. Data were acquired using a MEA1060-Inv-BC-Standard amplifier (MCS) with frequency limits of 5000 Hz and a sampling rate of 10 kHz per electrode mounted on an Olympus IX71 inverted microscope. Recordings were carried out under constant 37°C and 5% $CO_2$ conditions, identical to incubator conditions.

### Spike sorting

Raw data were filtered, offline, at 200 Hz using a Butterworth high-pass filter. Spikes cutouts were then detected, offline, using MC Rack software (MCS) based on a fixed threshold set to between 4–5 standard deviations from noise levels. In order to ensure the veracity of the detected spikes, TTX was added at the end of some experiments, resulting in an immediate loss of all spiking activity. Only the first 20 min

of each hour were used for all analyses in order to reduce computation time. We showed that 20 min can reliably represent the MFR of a full-hour by comparing the MFR of 792 20 min segments to the MFR of the full hours represented by those segments. We found that only 9.6% of segments were more than 10% different from their full hour and none were more than 20% different (*Figure 2—figure supplement 4A*). Furthermore, we calculated the coefficient of variation (CV) for 104 units over the course of 8 consecutive hr using different bin sizes. The CV represents the variability of MFRs within each unit over a given time-frame. We found that small bin sizes resulted in significantly higher CVs compared to 1 hr bins while bins larger than 10 min were not significantly different from bins of a full hour (CV = 121.3% ± 8.4, 53.4% ± 4.1, 25.1% ± 2, 19.2% ± 1.4, 14.8% ± 1.1, 13.5% ± 1, 11.60% ± 0.8, for bins of 10 s, 1, 5, 10, 20, 30, and 60 min, respectively; *Figure 2—figure supplement 4B*). This is an indication that, while there is always some intrinsic variability in per unit MFR, 20 min is an accurate representation of a full hour.

Spike cutouts were then transferred to Offline Sorter (Plexon Inc., Dallas, Texas, USA) for spike sorting. Spikes were plotted in 2-D or 3-D principal component (PC) space and unit clusters were semi-automatically detected using K-means clustering algorithm followed by template sorting. Clusters were then manually inspected to insure stability throughout experiment. Only clusters that fulfilled the following requirements were considered units and used for analysis: (1) there was no spiking during the absolute refractory period. (2) The clusters were well defined relative to other clusters from the same electrode throughout the entire experiment. (3) There were no sudden jumps in cluster location on PC axes. (4) The cluster is not centered around the origin of the PC axes. (5) Auto-correlation histograms showed a distinct peak at $t \neq 0$. A cluster was classified as 'multiunit' if the autocorrelogram lacked a clear refractory period.

## MEA data analysis

All analysis was performed using custom-written scripts in MATLAB (Mathworks, Natick, Massachusetts, USA). Network mean firing rates were calculated by averaging the mean firing rates of all units for a given time-point. For comparison of single unit parameters, we checked whether the difference in values between two time-points for a given unit significantly deviated from a null (zero centered) distribution using a bootstrapping method (see 'bootstrap.m' in *Source code 1*). The two time segments to be compared were divided into 1 min bins which were then randomly shuffled 10,000 times into two groups. The differences between the means of the two randomly shuffled groups produced a null-distribution. The real difference was significant if it fell outside of the 95% confidence interval of the null-distribution. The length of time segments, size of bins, and number of iterations had no effect on bootstrapping results (*Figure 2—figure supplement 4C–E*).

## Burst detection

For population-burst detection and analysis, we used the following algorithm (see 'network_burst_analysis.m' in *Source code 1*). We start with a set of spike time sequences acquired from a group of single units. The sequences are then projected onto a single timeline to produce an ordered time sequence.

$$\Theta = \bigcup_{\alpha, i} \left\{ t_i^\alpha \right\}$$

Here, $t_i^\alpha$ is the $i$th spike inside the $\alpha$th unit sequence.

Next, we estimate the local spike density $\hat{f}(t)$. To reduce binning issues, we use Parzen kernel density estimation (KDE) approach (*Parzen, 1962*):

$$\hat{f}(t) = \sum_i \rho_\sigma (t - t_i),$$

where we choose a Gaussian kernel.

$$\rho_\sigma (t) = \left( 2\pi\sigma^2 \right)^{-1/2} \exp(-t^2 / 2\sigma^2).$$

The timeline is discretized with sampling intervals of 50 ms; for the kernel, we set $\sigma^2 = 0.5$.

The resulting spike density function peaks strongly around potential burst locations, where the spike density is especially high. We use this fact as a cue to detect network bursts—every peak higher than a certain threshold is considered a candidate burst. Spike density threshold is estimated as the

95% percentile for a random Poisson process producing the same quantity of spikes. The timelines for each unit and thus for the entire network are divided into hourly intervals; we assume that throughout each interval the physiological state of the network and thus the spike statistics do not change. Thus, a different threshold is calculated for each hourly interval. Since we are looking for collective behavior modes, as a concluding filtering stage, we eliminate candidates generated by a small number of units. A unit is considered active if during the candidate burst it contributed at least a predefined number of spikes (usually 2–3). Then, for each hourly interval, a histogram of number of active units in a burst is calculated. The threshold for candidate burst elimination is set at the predefined percentile. In practice, choosing any value between 20–30% produces very similar results.

For detection of bursts at the single-unit level, bursts were defined as 3 or more spikes at a minimum of 50 Hz (see 'single_unit_bursts.m' in *Source code 1*). We analyzed data for all the combinations of parameters ranging from a minimum of 2, 3, and 4 spikes at frequencies of ≥20, 50, and 100 Hz and found no difference in the qualitative results showing that this analysis is robust over a wide range of definitions (*Figure 4—figure supplement 2*).

## Calcium imaging

Hippocampal cultures were infected by AAV2/1-Syn-GCaMP6f at 5–8 day in vitro. The experiments were performed 8–12 days post-infection when expression reached stable levels. Time-lapse images were acquired at 37°C and $CO_2$ controlled environment, using Nikon Eclipse Ti microscope with air 20× objective lens (NA = 0.45), controlled via iQ software (Andor, UK). Time-lapse images were collected at 20 Hz (1920 × 1080 pixels; 624 × 351 μm) using Neo sCMOS camera (Andor) and binned 4 × 4 pixels. The light source was AMH-2000 metal-halide lamp. The excitation wavelength was 485/20 nm, the emission band-pass filter was 525/50 nm. Imaging parameters were optimized to minimize photobleaching and phototoxicity, while preserving sufficient signal-to-noise ratio and temporal resolution. Light intensity was kept constant in all measurements during the 2 days of the experiment. Images were taken each for 2 min with 5 min intervals for 10 cycles, thus sampling 20 min of activity over a 65 min period. We saw no discernible changes in neurons' morphologies or in mean $Ca^{2+}$-transient amplitudes within or across sessions under our imaging conditions (*Figure 3—figure supplement 1D,E*). For AP-evoked signals (*Figure 3—figure supplement 1A,B*), synaptic blockers (50 μM AP5 and 20 μM CNQX) were used to block recurrent activity.

Signal intensity was measured using imageJ software by selecting ROIs over the soma of neurons expressing GCaMP6f while avoiding the nucleus. The fluorescence time course of each cell was measured by averaging all pixels within the ROI. Only cells that were in the same estimated focal plane and visually separated from neighboring cells and surrounding neuropil were used. For long-term GCaMP6f imaging, baseline fluorescence images of multiple sessions were inspected manually, and only the cells that could be clearly identified in all imaged sessions in the focal plane were included in the analysis. $\Delta F/F$ was quantified for each cell as change in fluorescence divided by baseline fluorescence measured 1 s before the spontaneous signal or stimulation. The sum of amplitudes of all the events was divided by the recording time to estimate the rate of $Ca^{2+}$ transients.

## Whole-cell recordings in hippocampal culture

Experiments were performed at room temperature in a recording chamber on the stage of FV300 inverted confocal microscope (Olympus, Japan). Extracellular Tyrode solution contained (in mM): NaCl, 145; KCl, 3; glucose, 15; HEPES, 10; $MgCl_2$, 1.2; $CaCl_2$, 1.2; pH adjusted to 7.4 with NaOH. Whole-cell patches were recorded using the following intracellular solution (in mM): Cs-MeSO$_3$, 120; HEPES, 10; NaCl, 10; $CaCl_2$, 0.5; $Mg^{2+}$–ATP, 2; $Na_3$GTP, 0.3; EGTA, 10 for mEPSC and E/I experiments; pH adjusted to 7.25 with NaOH. Serial resistance was not compensated. For mEPSCs recordings, Tetrodotoxin (TTX; 1 μM), amino-phosphonopentanoate (AP-5; 50 μM), and gabazine (30 μM) were added to the Tyrode solution. For measurement of excitation/inhibition (E/I) balance, sEPSCs and sIPSCs were isolated in the same cell based on reversal potentials of $GABA_A$R-mediated and AMPAR-mediated currents, respectively. When corrected for the liquid junction potential, the reversal potential for E ($V_E$) was close to 10 mV and $V_I$ was close to −65 mV, close to the predicted value for the intracellular and extracellular solutions used in the present study. For intrinsic excitability measurements, the following intracellular solution was used (in mM): K-gluconate 120; KCl 10; HEPEs 10; Na-phosphocreatine 10; ATP-Na$_2$ 4; GTP-Na 0.3; $MgCl_2$ 0.5. Recordings were done in the presence of synaptic blockers (in μM: 25 DNQX, 50 APV, and 10 bicuculline). Frequency vs current intensity curves were plotted

by measuring the average rate of action potentials in current-clamp during 500-ms long depolarizing steps of increasing intensity; a small DC current was injected to maintain membrane potential at −65 mV in between depolarizations. Input resistance ($R_{in}$) was measured by calculating the slope of the voltage change in response to increasing current injections from −80 pA to −20 pA in 20 pA increments. Access resistance was between 5–15 MΩ. Neurons were excluded from the analysis if RMP was >−55 mV, serial resistance was >15 MΩ, and $R_{in}$ was <80 MΩ or if any of these parameters changed by >20% during the recording. Signals were recorded using a MultiClamp 700B amplifier, digitized by DigiData1440A (Molecular Devices, Sunnyvale, California, USA) at 10 kHz, and filtered at 2 kHz. Electrophysiological data were analyzed using MiniAnalysis (Synaptosoft, Decatur, Georgia, USA) for mEPSCs and in pClamp10 (Molecular Devices) for sEPSCs and sIPSCs. The integrated conductances $G_E$ and $G_I$ were calculated according to the following equations $G_E = \int_0^t \frac{sEPSC}{t\left(V_M - V_{Erev}\right)}$ and $G_I = \int_0^t \frac{sIPSC}{t\left(V_M - V_{Irev}\right)}$.

## FM dye imaging and analysis

Activity-dependent FM1-43 styryl dye was used to estimate basal synaptic vesicle recycling and exocytosis. Action potentials were elicited by passing 50 mA constant current for 1 ms (~50% above the threshold for eliciting action potential) through two platinum wires, separated by ~7 mm, and close to the surface of the coverslip. The extracellular Tyrode solution contained non-selective antagonist of ionotropic glutamate receptors (kynurenic acid, 0.5 mM) to block recurrent neuronal activity. Synaptic vesicles were loaded with 10 µM FM1-43. FM loading and unloading were done using protocols described previously (*Abramov et al., 2009*). The fluorescence of individual synapses was determined from the difference between images obtained after staining and after destaining (Δ*F*). To estimate vesicle recycling/release during low frequency stimulation, we quantified: (i) Δ*F* signal for staining by 30 action potentials at a rate of 1 Hz stimulation; (ii) FM destaining rate during 1 Hz stimulation following staining of boutons by maximal stimulation. For FM-(+) puncta detection, Δ*F* images have been analyzed (only the puncta exhibiting ≥90% destaining were subjected to analysis). Detection of signals has been done as described (*Abramov et al., 2009*).

## Chemical reagents

FM1-43 and Advasep-7 were purchased from Biotium (Hayward, California, USA), TTX from Alamone Labs (Israel), baclofen, CNQX, and AP-5 from Tocris (UK).

## Statistical analysis

Error bars shown in the figures represent standard error of the mean (SEM). The number of experiments is defined by n. Student's paired *t*-tests were used in all the experiments where the effect of baclofen was tested in the same cell/synapse (\*p < 0.05; \*\*p < 0.01; \*\*\*p < 0.001), unless otherwise noted. Unpaired t-tests were used to compare different populations of synapses. One-way ANOVA Kruskal–Wallis nonparametric test was used to compare several populations of synapses. Nonparametric Spearman's test has been used for correlation analysis. For comparison of mEPSC amplitude or frequency under different conditions, 200 mEPSCs were randomly selected for each cell and pooled for each condition. A Kolmogorov–Smirnov (K–S) test was used to compute differences in distributions across the pooled datasets.

## Acknowledgements

We thank Ayal Lavi for his help with establishing MEA experiments, Eitan Zahavi and Eran Perlson for valuable technical help and access to the microscope with stage incubator, Eran Stark for comments on the manuscript and suggestions on spike sorting, Ronnie Maor and Gal Chechik for the development of the early version of burst detection algorithm, and all the laboratory members for discussions.

## Additional information

### Funding

| Funder | Grant reference number | Author |
|---|---|---|
| European Research Council | 281403 | Inna Slutsky |

| Funder | Grant reference number | Author |
|---|---|---|
| United States–Israel Binational Science Foundation | 2007199 | Inna Slutsky |
| Israel Science Foundation | 993/08, 170/08 and 398/13 | Inna Slutsky |
| Legacy Heritage Fund | 865/11 | Inna Slutsky |

The funders had no role in study design, data collection and interpretation, or the decision to submit the work for publication.

#### Author contributions

ES, BS, Conception and design, Acquisition of data, Analysis and interpretation of data, Drafting or revising the article; IV, HM-P, Acquisition of data, Analysis and interpretation of data; IN, MS, Analysis and interpretation of data, Drafting or revising the article; IS, Conception and design, Analysis and interpretation of data, Drafting or revising the article

#### Author ORCIDs

Edden Slomowitz, http://orcid.org/0000-0003-0549-6538

#### Ethics

Animal experimentation: All animal experiments were approved by the Tel Aviv University Committee on Animal Care (permit number M-11-005).

## Additional files

#### Source code

• Source code 1. Contains MATLAB software and code.

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
