## [Decision Letter]

Thank you for sending your work entitled “Interplay between population firing stability and single neuron dynamics in hippocampal networks” for consideration at *eLife*. Your article has been favorably evaluated by Eve Marder (Senior editor) and 3 reviewers, one of whom, Sacha Nelson, is a member of our Board of Reviewing Editors.

The Reviewing editor and the other reviewers discussed their comments before we reached this decision, and the Reviewing editor has assembled the following comments to help you prepare a revised submission.

The reviewers agreed that the issue of whether or not firing rate homeostasis occurs primarily in individual neurons or at the network level is highly important and that the findings in your study may force revision of the dominant view that homeostasis is primarily cell autonomous. They also agreed that the data are of high quality. However several points need clarifying or shoring up.

The major concerns are primarily technical:

1) Some of the reviewers were concerned about how exactly firing rate variability was quantified and what it reflects. The text states that “time-points to be compared were divided into one minute bins”, but does not make clear the intervals used: was it an hour divided into minutes or some other interval? The key question is whether individual neurons are tending to different rates over long time scales (the implied interpretation) or are moving randomly up and down over shorter time scales (could even be hours) such that repeated sampling finds them in different states.

2) There is also a technical concern over the stability of the recordings. It is known that neurons in culture can move, thus potentially changing the set of cells sampled at each electrode. More should be done to convince the reader that sufficient stability has been achieved to rule out the possibility that variability reflects variability of the neurons recorded. The finding that multi-units also show variability is not a sufficient control for this.

3) The authors demonstrate that baclofen is still effective after two days but have they ruled out the possibility that the relevant G-protein signaling is highly desensitized? One way to address this is to monitor activity during the washout period. The prediction is that activity should rebound. This would then offer the opportunity to observe bi-directional changes in activity of single neurons to further demonstrate the concordance, or lack thereof, of single neuron and network activity. It would also be useful to do this for the CNQX experiment.

In addition, the reviewers felt that the Introduction and Discussion could be shortened and some hyperbole reduced. For example, appeals to more “physiological” manipulations may not be warranted, since it is not clear that massive GABA-B inhibition is physiological. It is also not correct that prior vertebrate in vitro studies have not looked at network activity, while it is certainly true that these studies did not monitor activity of individual neurons over time.

---

## [Author Response]

*1) Some of the reviewers were concerned about how exactly firing rate variability was quantified and what it reflects. The text states that “time-points to be compared were divided into one minute bins”, but does not make clear the intervals used: was it an hour divided into minutes or some other interval? The key question is whether individual neurons are tending to different rates over long time scales (the implied interpretation) or are moving randomly up and down over shorter time scales (could even be hours) such that repeated sampling finds them in different states*.

We regret that the issue of time segments used was not clear. These issues are now addressed under the “Spike sorting” header of the Materials and methods section and in Figure 2—figure supplement 4. All analyses of MEA recordings were performed using the first 20 minutes of each hour of recording. This was done because of the large volume of data collected during each experiment. In order to ensure that 20 minutes is a reliable representation of a full hour we compared the mean firing rate (MFR) of 792 20-minute segments with the full hour that they represent. We found that on average, the 20-minute average segment was only 5.1 ± 0.4% different from the full-hour average. Out of the 792 segments analyzed, 90% were within 10% and none was more than 20% different from the corresponding full hour average (Figure 2—figure supplement 4).

In order to determine the statistical significance of the difference in MFR between two time-points we used a bootstrapping method. This was done by dividing the two time segments to be compared into bins of equal duration (for example, all data in the paper uses 20 minute time segments that were divided into 1 minute bins for bootstrapping analysis). MFR for each bin was then calculated. The bins were then randomized into two equal groups and the difference between the means of the randomized groups was calculated. The randomization was repeated 10,000 times to produce a null-distribution. The real MFR difference was significant only if it fell outside of the 95% confidence interval of the null distribution. To further ensure the validity of our data, we performed bootstrapping analyses of MFR while changing the different variables of the analysis. First, we tested whether the length of the time segment analyzed influenced the results. If 20 minute time segments are not sufficiently long to accurately represent the MFR over one-hour segments, increasing the duration of the time segments should change the results of the bootstrapping analysis. We saw no changes when performing bootstrapping analysis using time segments of 20 minutes, 1 hour or 2 hours (Figure 2—figure supplement 4) and, therefore, conclude that our results are not due to sampling errors. Likewise, we ran the bootstrapping analysis while changing the bin size and the number of iterations used to create the null-distribution. The results were extremely stable to changes in these parameters (Figure 2—figure supplement 4). We therefore conclude that our bootstrapping analysis is robust and represents true differences between MFRs of different time-points.

In order to further assess the minimal time required to avoid sampling errors we divided 8 consecutive hours of recording into time segments ranging from 10 seconds to 1 hour. We then calculated for each unit the coefficient of variation (CV) of the MFR for each segment size. Since CV assesses the variability of a given sample, we expected to reach a plateau for segment sizes long enough to reflect the intrinsic variability of the units. We found that the CV, when using time segments shorter than 10 minutes, indicated significantly higher variability compared to one-hour long segments. When increasing segment duration above 10 minutes, the CV stabilized (Figure 2—figure supplement 4). From this, we conclude that 20 minutes is sufficient time to accurately represent the basal, intrinsic, firing rate variability while allowing us to make comparisons to other time-points.

*2) There is also a technical concern over the stability of the recordings. It is known that neurons in culture can move, thus potentially changing the set of cells sampled at each electrode. More should be done to convince the reader that sufficient stability has been achieved to rule out the possibility that variability reflects variability of the neurons recorded. The finding that multi-units also show variability is not a sufficient control for this*.

A) In order to ensure stability of recorded populations we employed two complementary methods:

First, we acquired images both at the beginning and following 2 days of consecutive recording. In order to quantify the movement of single neurons we marked cells that appear well isolated and that could be reliably identified in both images. We then created a composite image using pairwise stitching in ImageJ (Preibisch et al., 2009) using the immobile electrodes as a reference points and measured cell-by-cell movement. In order to establish the accuracy of our measurements we repeated the measurement of 11 cells 10 times to establish the standard deviation of our measurement technique. The mean SD of the 11 cells was 0.29 ± 0.02 µm (Figure 10), showing that the measurements have a submicron level of accuracy. We next measured the movements of 100 cells and found a mean movement of 1.6 ± 0.1 µm (Figure 10). This is exceedingly small compared to the size of the neurons (11.4 ± 0.27 µm) and the 30 µM diameter of the recording electrode and therefore should not have a significant effect on the population recorded (Figure 10).Author response image 1.

Second, since spike waveforms are dependent, among other things, on the distance of the cell from the recording electrode (Nam et al., 2011), we would expect to see changes in waveform shape over time if there is significant movement. While performing spike sorting we indeed saw small drifts of waveform shape. However, in order to ensure the population stability, we only used units whose waveform remained highly separated throughout the recording. Units whose waveforms suddenly appeared or disappeared during the recording were not used for analysis.

The combination of these two methods, showing both physical and electrical evidence of population stability, suggest that the single-unit variability we observed was not contaminated by analysis of different sets of cells to any significant degree.

B) We used a different methodology—somatic calcium imaging—to directly address this question in visually identified neurons. We chose genetically encoded Ca^2+^ indicator GCaMP6f that displays single-action-potential sensitivity and broad dynamic range (5). We showed that the Δ*F*/*F* amplitude of calcium transients is proportional to the number of spikes that triggered it (Figure 3—figure supplement 1). To examine the stability of single neurons and neuronal populations at extended timescales, we performed time-lapse imaging of somatic Ca^2+^ dynamics during a baseline recording period, 10 minutes and 2 days following baclofen application. Our results show that while Ca^2+^ dynamics at the population level was stable, 54% of neurons did not return to their baseline values following 2 days baclofen incubation. Thus, calcium imaging in visually identified neurons confirms electrophysiological data, strengthening our conclusion on stabilization of population dynamics despite imprecise homeostatic compensations of single neurons.

*3) The authors demonstrate that baclofen is still effective after two days but have they ruled out the possibility that the relevant G-protein signaling is highly desensitized? One way to address this is to monitor activity during the washout period. The prediction is that activity should rebound. This would then offer the opportunity to observe bi-directional changes in activity of single neurons to further demonstrate the concordance, or lack thereof, of single neuron and network activity. It would also be useful to do this for the CNQX experiment*.

In order to fully rule out desensitization of G-protein signaling following prolonged exposure to baclofen, we recorded MEAs for 2 days in the presence of baclofen and subsequently washed the baclofen out of the recording chamber. We observed a significant increase in MFR to approximately 300% of MFR during both baseline and 2 day after baclofen addition (relative to baseline: 306% ± 67; relative to 2 days baclofen: 296% ± 38). This increase was observed at both the single unit and the network level (Figure 2—figure supplement 1).

Additionally, we saw no change in presynaptic sensitivity to baclofen, as measured using FM-dyes, after 2 days of baclofen incubation (Figure 2—figure supplement 1).

Likewise, CNQX washout caused a significant increase in MFR to 212 ± 28% relative to baseline MFR and to 442 ± 67% relative to the MFR after two days of CNQX (Figure 9—figure supplement 2).

We therefore conclude that both GABA_B_R-mediated G-protein signaling and AMPARs remain sensitive following two day exposure to baclofen and CNQX, respectively.

*In addition, the reviewers felt that the Introduction and Discussion could be shortened and some hyperbole reduced. For example, appeals to more “physiological” manipulations may not be warranted, since it is not clear that massive GABA-B inhibition is physiological. It is also not correct that prior vertebrate in vitro studies have not looked at network activity, while it is certainly true that these studies did not monitor activity of individual neurons over time*.

Thanks, we followed these suggestions.